# Experiments on Rare-Earth Element Extractions from Umber Ores for Optimizing the Grinding Process

**Yutaro Takaya** [1,2,3,4,*], **Meiqi Wang** [1], **Koichiro Fujinaga** [2,3], **Etsuo Uchida** [1], **Tatsuo Nozaki** [2,3,4,5] **and Yasuhiro Kato** [2,3,4]

1   Department of Resources and Environmental Engineering, School of Creative Science and Engineering, Waseda University, 3-4-1 Okubo, Shinjuku-ku, Tokyo 169-8555, Japan; wangmeiqi@akane.waseda.jp (M.W.); weuchida@waseda.jp (E.U.)
2   Ocean Resources Research Center for Next Generation, Chiba Institute of Technology, 2-17-1 Tsudanuma, Narashino, Chiba 275-0016, Japan; koichiro.fujinaga@p.chibakoudai.jp (K.F.); nozaki@jamstec.go.jp (T.N.); ykato@sys.t.u-tokyo.ac.jp (Y.K.)
3   Frontier Research Center for Energy and Resources, School of Engineering, the University of Tokyo, 7-3-1 Hongo, Bunkyo-ku, Tokyo 113-8656, Japan
4   Research and Development Center for Submarine Resources, Japan Agency for Marine-Earth Science and Technology (JAMSTEC), 2-15 Natsushima-cho, Yokosuka, Kanagawa 237-0061, Japan
5   Department of Planetology, Kobe University, 1-1 Rokkodai-cho, Nada-ku, Kobe, Hyogo 657-8501, Japan
*   Correspondence: y-takaya@aoni.waseda.jp; Tel.: +81-3-5286-3318

**Abstract:** Ancient hydrothermal metalliferous sediments (umber) have recently attracted attention as a new rare-earth element resource. We conducted chemical leaching experiments on three different umber ores to optimize the hydrometallurgical extraction process, especially regarding the grinding process. The three umber ore samples, which were collected from Japanese accretionary complexes (Kuminiyama and Aki umber) and Troodos ophiolite (Cyprus umber), had different chemical, mineral, and physical properties, and showed different leaching behaviors. The experimental results revealed that the physical properties (density and P-wave propagation velocity) principally controlled the extent of REY (lanthanides and yttrium) extraction from the umber ore samples, and REY extraction from umber samples clearly increased with the decrease in the density and P-wave propagation velocity. The differences in physical properties of the umber samples are attributable to the pressure and thermal history of each ore sample, and it was revealed that umber samples which underwent strong metamorphism are not suitable for actual development. The results also suggested that the optimum particle size (optimum grinding level) of umber samples is simply predictable based on the physical properties. The results of this study should be valuable for future efforts to procure these important mineral resources.

**Keywords:** ferromanganese deposits; metalliferous sediment; chemical leaching; REY; accretionary complex

## 1. Introduction

Rare-earth elements (REE) is a collective term for 17 chemically similar elements (lanthanides, scandium, and yttrium). The term "REY" (lanthanides and yttrium), which was first defined in Kato et al. [1], is also used in this study because the geochemical behavior of Sc differs somewhat from that of REY. REE are essential materials for advanced industries, and the global demand for REE is predicted to increase rapidly in the next few decades [2,3]. Presently, China produces approximately 80% of the world's annual consumption of REE [4], and therefore, the exploration of new REE mines

including seafloor mineral resources with the aim of achieving a stable supply is of worldwide importance [5,6].

Strata-bound ferromanganese deposits on land, known as umber, have recently attracted attention as a new REE resource [7,8] and are considered as analogues to oxide-based REE deposits on land [9]. These deposits are of hydrothermal metalliferous sedimentary origin and consist mainly of hydroxides (mainly iron and manganese oxide/hydroxide), volcanic glass, and adsorbed elements (REE, P, Mo, and V) from ambient seawater [10–12], with low concentrations of radioactive elements (Th and U) also present; notably, radioactive elements are a source of serious environmental concern in many on-land REE deposits [13,14]. Umber deposits are considered to be closely related to mid-ocean ridge volcanism [15–17] and generally overlie basaltic rocks (or greenstones).

There are many umber deposits in Japanese accretionary complexes, and resource amount estimations as well as geochemical studies already have been conducted [7,8,18–21]. Umber deposits are also extensively exposed on land as part of ophiolites such as in Cyprus (Troodos ophiolite) and Oman (Semail ophiolite) [22–24]. Josso et al. [9] conducted REY extraction experiments on the Cyprus umber and discussed the optimum procedural conditions (acid concentration, temperature, solid-liquid ratio, and leaching time). However, the factors that control the leaching behavior of umber were not clarified in the previous studies. In this study, therefore, we conducted a series of chemical leaching experiments using three different umber samples collected from Japanese accretionary complexes and Troodos ophiolite to elucidate the factors affecting the extraction behaviors of REY. In addition, we have established a simple prediction method for REY extraction efficiency from umber ores targeted for future development.

## 2. Materials and Methods

### 2.1. Samples

We used three umber samples in the experiments to clarify the chemical/physical factors that could affect the leaching behavior. Two samples (Kunimiyama and Aki Umbers) were taken from central Shikoku, southwestern Japan, and the other sample was from Cyprus (Troodos ophiolite). The constituent minerals of each sample were determined by X-ray powder diffraction (XRD, RINT-ULTIMA III, Rigaku Corporation installed at the University of Tokyo, Tokyo, Japan) and PDXL (integrated XRD software).

Kunimiyama umber: The Kunimiyama deposit in central Shikoku is one of the largest stratiform ferromanganese deposits in Japan [25–28]. This sample was collected from Kagami-mura (the Kunimiyama area) in the northwestern part of Kochi City, southwestern Japan, which belongs to the Northern Chichibu Belt. The geology of this area consists of greenstone, chert, mélange, and sandstone. The age of the Sumaizuku Unit, including the Kunimiyama area, of the Northern Chichibu Belt is estimated to correspond to the Carboniferous to Middle Jurassic periods [29,30]. In addition, the age of radiolarian fossils from red chert, which occurs above the ferromanganese deposit, corresponds to the early Permian [31]. This umber sample exhibited a reddish-brown to dark reddish-brown color. Thin (0.05–0.1 mm in width) calcite veins were rarely observed in this sample. Hematite ($Fe_2O_3$) and bementite ($Mn_8Si_6O_{15}(OH)_{10}$) were found to be the main constituent minerals of this sample (Figure 1a). Weak peaks of rare-earth oxide were also detected by the XRD analysis, but these data are disputable because the total amount of REY (ΣREY) of the Kunimiyama umber was less than 0.2 wt. %.

Aki umber: This sample was taken from the Aki ferromanganese deposit in the Aki Group belonging to the Northern Shimanto Belt in the northern part of Aki City, in the western part of Kochi Prefecture, southwestern Japan. The Aki Group consists of the Tei-mélange and Tsukimiyama-mélange, and this sample was collected from the Tei-mélange [8]. The sedimentary age of the Tei-mélange as determined by radiolarian fossils corresponds to the Albian to Cenomanian (112–90.4 Ma: Middle Cretaceous) for the bedded chert, Turonian to Santonian (90.4–80.3 Ma) for the shale overlying the chert, and Campanian (83–74 Ma) for the mudstone [32]. This sample exhibited a reddish-brown to

dark reddish-brown color. Quartz ($SiO_2$) and hematite were found to be the main constituent minerals of this sample (Figure 1b).

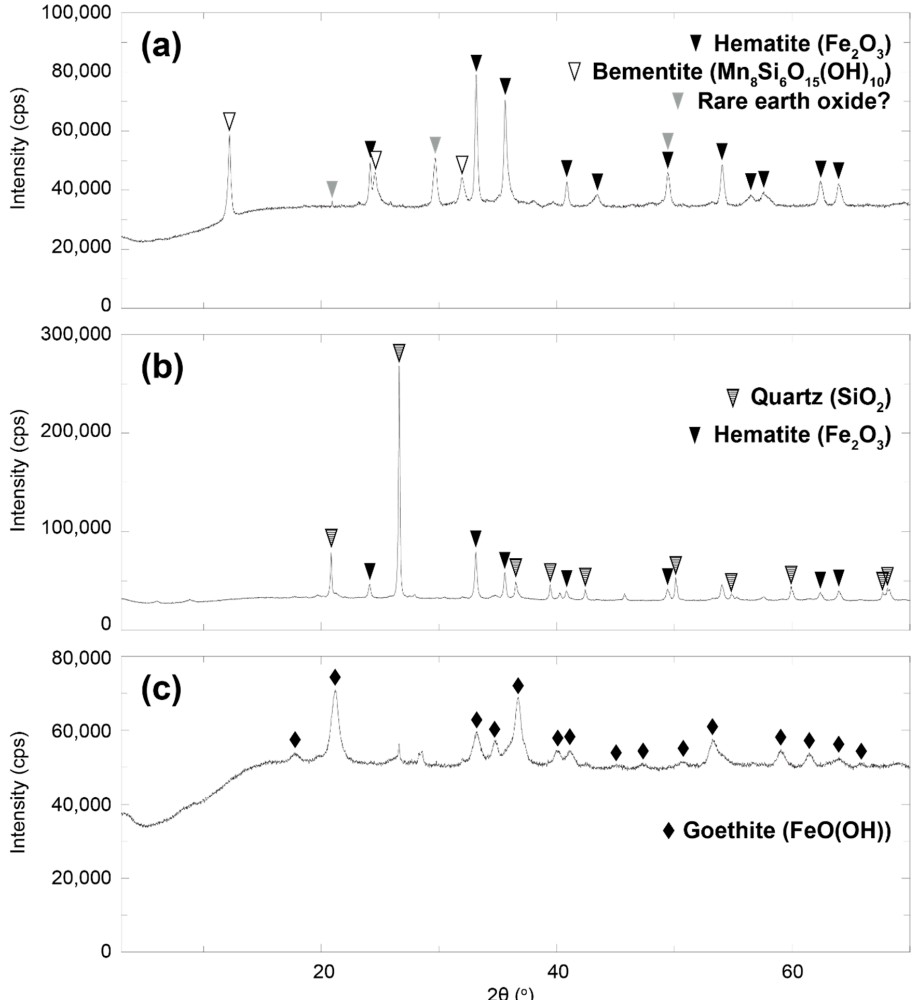

**Figure 1.** X-ray diffractograms of umber samples. (**a**) Kunimiyama umber, (**b**) Aki umber, and (**c**) Cyprus umber.

Cyprus umber: This sample was collected from the Kampia area in the central part of Cyprus Island, on the east side of the Mediterranean. The Kampia area is located at the northern border of the Troodos ophiolitic massif (mountains) and is near the Theotokos Monastery and Kampia sulfide mine [33,34]. The Troodos massif is a part of the Tethyan oceanic lithosphere and is composed of a complete sequence of oceanic crust (pelagic sediments, pillow lavas, sheeted dikes, felsic to ultramafic plutonic rocks, and mantle rocks) [22,35]. Intense uplift of the Troodos ophiolitic massif was suggested to have started in the Late Pliocene to Early Pleistocene based on a paleo-environmental study of the Athalassa Formation [36]. Umber lies on basaltic lavas, some of which are accompanied by cherts and limestone at the top [34]. Although umber is largely exposed on land, its thickness is often less than 1 m [9,34]. This sample exhibited a homogenous brown color. The XRD analysis revealed that the constituent mineral was goethite ($FeO(OH)$) (Figure 1c).

## 2.2. Experiments

### 2.2.1. Sample Preparation

First, the umber samples were cut into chips that were 1 cm thick. After polishing the surface of chip samples by 70 μm mesh abrasive-coated paper, the chip samples were ultrasonically cleaned

in Milli-Q water for at least 10 min. After drying, the samples were crushed in an agate mortar and sieved into four size fractions (25–150, 150–1000, 1000–2000, and 2000–4000 µm). Particles smaller than each size fraction were removed by ultrasonic cleansing in ethanol on the sieve for 10 min. The sieved samples were dried and used in the chemical leaching experiments.

### 2.2.2. Whole Rock Chemical Analyses

Whole rock chemical analyses of the sieved samples (in total, 12 samples; three samples for each of the four size fractions) were conducted by inductively coupled plasma quadrupole mass spectrometry (ICP-QMS) (iCAP Q: Thermo Fisher Scientific instrument installed at the University of Tokyo). A dried 0.05 g sample was digested with a solution consisting of 0.8 mL $HClO_4$, 2 mL HF, and 4 mL $HNO_3$ at 130 °C for 2 h. After this mixed acid solution was dried by stepwise heating, 1.5 mL HCl and 0.5 mL $HNO_3$ were added and the solution was heated at 90 °C for 3 h. The mixed acid solution was then dried by stepwise heating up to 160 °C, and a 10 mL mixed acid solution (2 wt. %, $HNO_3$:HCl:HF = 20:5:1) was added to this dried sample and again the sample was heated at 90 °C for 3 h. Finally, the sample solution was diluted with a 2 wt. % mixed acid solution and used for the ICP-QMS analysis. For only the 2000–4000 µm fraction, we used 0.1 g samples and doubled the amount of acid solution for the analyses because it was difficult to adjust the sample weight to 0.05 g.

In the analyses, we prepared three different specimens for each umber sample to examine the sample heterogeneity. Therefore, we prepared 36 different solutions (three samples × four size fractions × three specimens) and calculated the standard deviation for each sample (Tables 1–3).

### 2.2.3. Chemical Leaching Experiments

In this study, hydrochloric acid and sulfuric acid were used as a leachate solution with reference to previous studies on REE leaching for deep-sea sediments [37,38] and umber samples [9]. The acid concentration was set to 0.5 mol/L for hydrochloric acid and 0.25 mol/L for sulfuric acid, and the leaching temperature was set to 25 °C. The leaching time was set to 5, 30, 180, 720, and 1440 min. A total of 120 experiments were conducted to cover all combinations of these conditions (three samples, four particle sizes, five leaching times, and two kinds of leachate).

The experimental procedure is shown in Figure 2. Umber samples were dried at 60 °C for over 24 h prior to the leaching experiments. A sample weighing 0.2 ± 0.002 g was combined with 3 mL of the leachate in a PTFE (polytetrafluoroethylene) vessel. We shook the vessel to mix the umber samples with leachate and kept it at a temperature of 25 °C. After the prescribed leaching time, we again shook the vessel and collected the leaching solution. The solution was immediately filtered with a 0.45 µm mesh membrane filter, the pH was measured, and then, the sample was diluted 100 times with a mixed acid solution (2 wt. %, $HNO_3$:HCl:HF = 20:5:1) for the analysis by ICP-QMS.

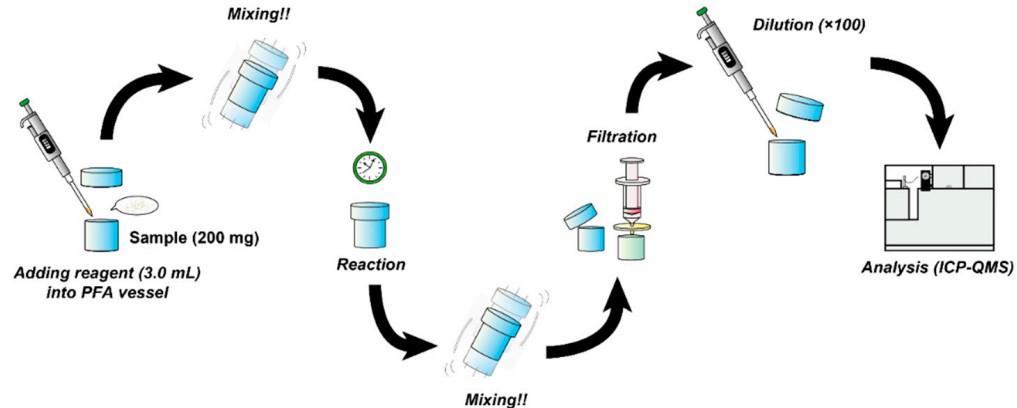

**Figure 2.** Schematic diagram of the procedure for the leaching experiment.

**Table 1.** Chemical composition of the Kunimiyama umber determined by ICP-QMS.

| Particle Size [μm] | 25–150 | | 150–1000 | | 1000–2000 | | 2000–4000 | |
|---|---|---|---|---|---|---|---|---|
| | Average | 1δ | Average | 1δ | Average | 1δ | Average | 1δ |
| Na (%) | 0.01 | 0.00 | 0.01 | 0.00 | 0.01 | 0.00 | 0.01 | 0.00 |
| Mg | 1.10 | 0.04 | 1.26 | 0.04 | 1.25 | 0.06 | 1.23 | 0.10 |
| Al | 1.08 | 0.07 | 1.29 | 0.03 | 1.23 | 0.05 | 1.09 | 0.00 |
| P | 0.39 | 0.01 | 0.38 | 0.08 | 0.42 | 0.13 | 0.38 | 0.22 |
| K | 0.00 | 0.00 | 0.00 | 0.00 | 0.00 | 0.00 | 0.00 | 0.00 |
| Ca | 3.17 | 0.14 | 2.40 | 0.19 | 1.76 | 0.35 | 1.85 | 0.79 |
| Sc (ppm) | 21.89 | 0.70 | 24.46 | 0.22 | 25.10 | 0.71 | 25.46 | 2.45 |
| Ti (%) | 0.30 | 0.02 | 0.35 | 0.02 | 0.37 | 0.01 | 0.35 | 0.01 |
| V (ppm) | 1206.40 | 39.04 | 1299.16 | 25.90 | 1308.36 | 61.25 | 1196.71 | 54.74 |
| Cr | 20.80 | 0.75 | 21.66 | 0.81 | 24.38 | 3.77 | 23.52 | 3.31 |
| Mn (%) | 18.01 | 0.58 | 18.88 | 0.38 | 18.71 | 0.76 | 17.99 | 0.53 |
| Fe | 25.66 | 0.85 | 28.43 | 0.15 | 28.59 | 0.57 | 27.22 | 1.31 |
| Co (ppm) | 287.57 | 7.09 | 313.46 | 5.87 | 317.28 | 11.57 | 316.08 | 12.10 |
| Ni | 1408.46 | 30.48 | 1522.99 | 16.40 | 1542.80 | 64.20 | 1613.62 | 78.31 |
| Cu | 317.78 | 11.70 | 240.21 | 28.59 | 201.89 | 59.97 | 148.55 | 10.79 |
| Zn | 675.09 | 23.55 | 736.27 | 13.53 | 728.55 | 18.87 | 722.51 | 30.04 |
| As | 138.52 | 11.22 | 152.42 | 15.23 | 171.24 | 23.70 | 150.54 | 68.06 |
| Rb | 0.53 | 0.03 | 0.62 | 0.03 | 0.61 | 0.02 | 0.57 | 0.14 |
| Sr | 141.77 | 4.85 | 137.41 | 6.87 | 117.79 | 5.11 | 121.77 | 7.57 |
| Y | 240.16 | 8.71 | 243.56 | 11.30 | 273.22 | 22.34 | 258.28 | 69.84 |
| Zr | 344.77 | 20.16 | 421.93 | 73.88 | 419.31 | 27.38 | 397.43 | 57.84 |
| Nb | 35.54 | 1.01 | 38.91 | 0.16 | 40.86 | 1.64 | 38.61 | 3.64 |
| Mo | 273.92 | 34.30 | 99.17 | 49.43 | 48.78 | 8.36 | 49.22 | 10.73 |
| Cd | b.d.l. | - | b.d.l. | - | b.d.l. | - | b.d.l. | - |
| In | b.d.l. | - | b.d.l. | - | b.d.l. | - | b.d.l. | - |
| Sn | b.d.l. | - | b.d.l. | - | b.d.l. | - | b.d.l. | - |
| Sb | b.d.l. | - | b.d.l. | - | b.d.l. | - | b.d.l. | - |
| Cs | 0.43 | 0.01 | 0.44 | 0.01 | 0.43 | 0.02 | 0.47 | 0.05 |
| Ba | 247.32 | 10.77 | 274.72 | 40.81 | 200.42 | 7.87 | 203.92 | 96.32 |
| La | 397.37 | 11.97 | 421.66 | 9.04 | 429.54 | 17.23 | 418.61 | 14.85 |
| Ce | 241.47 | 8.12 | 259.51 | 5.63 | 256.43 | 3.02 | 252.38 | 15.96 |
| Pr | 93.55 | 3.13 | 97.92 | 2.31 | 100.85 | 3.54 | 97.91 | 6.61 |
| Nd | 378.95 | 10.68 | 387.80 | 10.26 | 403.44 | 13.29 | 391.54 | 34.37 |
| Sm | 83.53 | 2.47 | 84.48 | 2.33 | 88.35 | 4.27 | 84.94 | 10.37 |
| Eu | 19.16 | 0.59 | 19.17 | 0.51 | 20.13 | 1.20 | 19.49 | 2.88 |
| Gd | 81.04 | 2.38 | 81.01 | 2.80 | 85.24 | 5.89 | 84.23 | 15.77 |
| Tb | 12.32 | 0.43 | 12.37 | 0.40 | 13.08 | 0.84 | 12.90 | 2.37 |
| Dy | 70.71 | 2.51 | 71.65 | 2.23 | 75.82 | 4.58 | 74.77 | 14.41 |
| Ho | 13.34 | 0.46 | 13.43 | 0.35 | 14.18 | 0.80 | 13.81 | 2.75 |
| Er | 36.30 | 1.28 | 36.84 | 1.02 | 39.29 | 1.43 | 38.24 | 7.50 |
| Tm | 5.02 | 0.16 | 5.15 | 0.11 | 5.43 | 0.22 | 5.23 | 0.91 |
| Yb | 31.66 | 1.23 | 32.62 | 0.56 | 34.24 | 0.90 | 32.91 | 5.22 |
| Lu | 4.54 | 0.14 | 4.68 | 0.08 | 4.93 | 0.10 | 4.76 | 0.77 |
| Hf | 4.72 | 0.11 | 5.59 | 0.72 | 5.65 | 0.26 | 5.37 | 0.70 |
| Ta | 0.21 | 0.00 | 0.22 | 0.01 | 0.22 | 0.01 | 0.25 | 0.03 |
| W | 0.00 | 0.00 | 0.00 | 0.00 | 0.00 | 0.00 | 0.00 | 0.00 |
| Tl | b.d.l. | - | b.d.l. | - | b.d.l. | - | b.d.l. | - |
| Pb | 274.51 | 17.42 | 223.77 | 13.29 | 199.01 | 32.64 | 185.83 | 17.69 |
| Bi | 0.00 | 0.00 | 1.70 | 2.94 | 0.00 | 0.00 | 0.00 | 0.00 |
| Th | 8.83 | 0.30 | 9.93 | 0.69 | 10.14 | 0.60 | 9.76 | 1.10 |
| U | 1.42 | 0.03 | 1.52 | 0.03 | 1.59 | 0.12 | 1.52 | 0.16 |
| ΣREY | 1709.11 | 44.04 | 1771.85 | 32.44 | 1844.17 | 62.28 | 1790.01 | 144.75 |

b.d.l.: below the detection limit.

**Table 2.** Chemical composition of the Aki umber determined by ICP-QMS.

| Particle Size [μm] | 25–150 | | 150–1000 | | 1000–2000 | | 2000–4000 | |
|---|---|---|---|---|---|---|---|---|
| | Average | 1δ | Average | 1δ | Average | 1δ | Average | 1δ |
| Na (%) | 0.17 | 0.00 | 0.16 | 0.01 | 0.13 | 0.01 | 0.15 | 0.11 |
| Mg | 0.91 | 0.02 | 0.75 | 0.08 | 0.59 | 0.06 | 0.82 | 0.26 |
| Al | 2.52 | 0.07 | 2.14 | 0.19 | 1.70 | 0.11 | 2.21 | 1.27 |
| P | 0.35 | 0.01 | 0.27 | 0.03 | 0.19 | 0.01 | 0.31 | 0.30 |
| K | 1.03 | 0.03 | 0.95 | 0.07 | 0.68 | 0.09 | 0.87 | 0.65 |
| Ca | 1.17 | 0.07 | 0.80 | 0.07 | 0.58 | 0.09 | 0.90 | 0.42 |
| Sc (ppm) | 11.00 | 0.30 | 9.01 | 0.68 | 6.88 | 0.50 | 7.99 | 7.12 |
| Ti (%) | 0.13 | 0.01 | 0.11 | 0.01 | 0.08 | 0.01 | 0.12 | 0.12 |
| V (ppm) | 259.46 | 3.44 | 185.38 | 16.92 | 147.80 | 24.86 | 216.14 | 94.67 |
| Cr | 31.66 | 2.49 | 27.94 | 2.61 | 19.14 | 1.77 | 22.97 | 19.42 |
| Mn (%) | 3.35 | 0.09 | 2.20 | 0.26 | 1.54 | 0.22 | 2.05 | 0.78 |
| Fe | 17.45 | 0.31 | 15.52 | 1.09 | 10.96 | 0.77 | 14.26 | 12.22 |
| Co (ppm) | 40.40 | 0.94 | 29.37 | 3.05 | 22.12 | 2.51 | 32.36 | 11.38 |
| Ni | 127.35 | 3.02 | 90.89 | 11.97 | 66.29 | 6.54 | 93.90 | 43.10 |
| Cu | 536.53 | 24.44 | 342.47 | 53.91 | 505.02 | 61.53 | 291.14 | 199.91 |
| Zn | 297.09 | 6.62 | 214.14 | 24.67 | 153.16 | 17.94 | 235.77 | 90.92 |
| As | 98.64 | 1.16 | 69.73 | 5.19 | 56.61 | 5.31 | 81.73 | 42.95 |
| Rb | 51.78 | 0.73 | 49.12 | 3.79 | 35.39 | 6.10 | 46.81 | 36.81 |
| Sr | 123.28 | 2.50 | 90.33 | 9.17 | 68.74 | 3.78 | 86.46 | 36.17 |
| Y | 148.71 | 2.01 | 118.92 | 10.38 | 85.88 | 7.72 | 119.44 | 104.31 |
| Zr | 129.21 | 1.24 | 111.64 | 9.81 | 84.22 | 10.92 | 99.00 | 85.93 |
| Nb | 6.73 | 0.13 | 5.72 | 0.52 | 4.16 | 0.38 | 5.24 | 4.71 |
| Mo | 3.00 | 0.05 | 2.29 | 0.26 | 1.78 | 0.13 | 2.41 | 1.34 |
| Cd | b.d.l. | - | b.d.l. | - | b.d.l. | - | b.d.l. | - |
| In | b.d.l. | - | b.d.l. | - | b.d.l. | - | b.d.l. | - |
| Sn | b.d.l. | - | b.d.l. | - | b.d.l. | - | b.d.l. | - |
| Sb | b.d.l. | - | b.d.l. | - | b.d.l. | - | b.d.l. | - |
| Cs | 5.54 | 0.10 | 5.05 | 0.41 | 3.66 | 0.40 | 4.81 | 4.59 |
| Ba | 540.63 | 2.17 | 293.20 | 52.34 | 165.13 | 36.01 | 228.54 | 172.79 |
| La | 161.03 | 2.46 | 135.50 | 11.29 | 96.85 | 3.93 | 134.28 | 113.94 |
| Ce | 52.82 | 0.61 | 41.66 | 3.39 | 30.73 | 1.34 | 41.61 | 30.78 |
| Pr | 45.76 | 0.47 | 36.42 | 2.82 | 26.70 | 2.27 | 36.14 | 29.37 |
| Nd | 186.81 | 2.68 | 147.63 | 11.45 | 108.19 | 10.49 | 147.39 | 122.02 |
| Sm | 41.58 | 0.60 | 32.47 | 2.78 | 24.04 | 2.61 | 32.23 | 26.64 |
| Eu | 9.57 | 0.12 | 7.56 | 0.63 | 5.61 | 0.51 | 7.61 | 6.33 |
| Gd | 40.45 | 0.45 | 31.79 | 2.77 | 23.29 | 2.21 | 31.75 | 26.66 |
| Tb | 5.91 | 0.09 | 4.67 | 0.38 | 3.43 | 0.36 | 4.64 | 3.92 |
| Dy | 34.85 | 0.44 | 27.24 | 2.50 | 20.05 | 2.17 | 27.18 | 23.15 |
| Ho | 6.41 | 0.08 | 5.08 | 0.46 | 3.72 | 0.39 | 5.13 | 4.44 |
| Er | 17.74 | 0.17 | 14.01 | 1.23 | 10.13 | 0.97 | 14.09 | 12.33 |
| Tm | 2.44 | 0.02 | 1.91 | 0.16 | 1.40 | 0.13 | 1.92 | 1.68 |
| Yb | 14.97 | 0.19 | 11.73 | 1.06 | 8.46 | 0.75 | 11.82 | 10.18 |
| Lu | 2.07 | 0.03 | 1.62 | 0.15 | 1.16 | 0.11 | 1.63 | 1.42 |
| Hf | 1.75 | 0.02 | 1.57 | 0.08 | 1.11 | 0.07 | 1.29 | 1.13 |
| Ta | 0.31 | 0.01 | 0.28 | 0.00 | 0.19 | 0.01 | 0.23 | 0.21 |
| W | 0.00 | 0.00 | 0.00 | 0.00 | 0.00 | 0.00 | 0.00 | 0.00 |
| Tl | b.d.l. | - | b.d.l. | - | b.d.l. | - | b.d.l. | - |
| Pb | 89.04 | 2.16 | 72.57 | 5.93 | 56.62 | 6.79 | 67.43 | 51.82 |
| Bi | 0.00 | 0.00 | 0.00 | 0.00 | 1.72 | 2.98 | 1.80 | 3.12 |
| Th | 3.84 | 0.05 | 3.51 | 0.09 | 2.39 | 0.10 | 2.90 | 2.52 |
| U | 2.09 | 0.02 | 1.69 | 0.12 | 1.31 | 0.15 | 1.60 | 0.97 |
| ΣREY | 771.12 | 7.83 | 618.21 | 41.84 | 449.62 | 25.71 | 616.86 | 422.22 |

b.d.l.: below the detection limit.

**Table 3.** Chemical composition of the Cyprus umber determined by ICP-QMS.

| Particle Size [μm] | 25–150 | | 150–1000 | | 1000–2000 | | 2000–4000 | |
|---|---|---|---|---|---|---|---|---|
| | Average | 1δ | Average | 1δ | Average | 1δ | Average | 1δ |
| Na (%) | 0.10 | 0.00 | 0.11 | 0.01 | 0.10 | 0.01 | 0.11 | 0.01 |
| Mg | 0.94 | 0.01 | 0.94 | 0.06 | 0.91 | 0.02 | 1.01 | 0.01 |
| Al | 1.27 | 0.01 | 1.25 | 0.12 | 1.27 | 0.04 | 1.36 | 0.02 |
| P | 0.54 | 0.01 | 0.51 | 0.08 | 0.47 | 0.12 | 0.57 | 0.22 |
| K | 0.36 | 0.01 | 0.37 | 0.02 | 0.36 | 0.03 | 0.35 | 0.01 |
| Ca | 1.21 | 0.09 | 1.33 | 0.04 | 1.33 | 0.33 | 1.64 | 0.51 |
| Sc (ppm) | 6.41 | 0.08 | 6.38 | 0.35 | 6.50 | 0.25 | 6.47 | 0.41 |
| Ti (%) | 0.08 | 0.00 | 0.08 | 0.00 | 0.07 | 0.01 | 0.06 | 0.01 |
| V (ppm) | 839.23 | 6.53 | 843.49 | 38.25 | 822.70 | 63.74 | 806.32 | 8.71 |
| Cr | 28.07 | 3.01 | 17.22 | 0.65 | 14.94 | 1.47 | 14.84 | 0.99 |
| Mn (%) | 7.45 | 0.05 | 7.63 | 0.27 | 7.58 | 0.50 | 6.82 | 0.20 |
| Fe | 38.31 | 0.37 | 37.01 | 2.06 | 36.91 | 2.12 | 36.33 | 0.71 |
| Co (ppm) | 155.78 | 1.64 | 181.50 | 5.91 | 156.22 | 3.33 | 183.07 | 37.13 |
| Ni | 326.87 | 5.25 | 331.94 | 13.23 | 313.00 | 18.51 | 300.47 | 8.26 |
| Cu | 1079.80 | 8.74 | 1075.19 | 59.80 | 1056.16 | 53.53 | 1023.40 | 9.74 |
| Zn | 404.56 | 6.70 | 401.92 | 17.12 | 389.40 | 20.59 | 382.99 | 3.56 |
| As | 384.11 | 1.61 | 395.46 | 24.70 | 407.02 | 21.08 | 377.68 | 8.46 |
| Rb | 14.05 | 0.24 | 14.00 | 1.04 | 13.70 | 1.48 | 13.23 | 0.17 |
| Sr | 321.66 | 3.93 | 326.28 | 12.84 | 320.29 | 25.29 | 349.82 | 17.82 |
| Y | 105.38 | 1.80 | 97.79 | 6.30 | 93.99 | 16.21 | 116.15 | 26.68 |
| Zr | 73.06 | 0.67 | 71.06 | 4.96 | 70.18 | 4.40 | 70.03 | 0.80 |
| Nb | 2.99 | 0.02 | 2.95 | 0.23 | 2.87 | 0.17 | 2.91 | 0.02 |
| Mo | 12.65 | 0.08 | 13.25 | 0.47 | 12.03 | 0.71 | 12.72 | 0.76 |
| Cd | - | - | - | - | - | - | - | - |
| In | - | - | - | - | - | - | - | - |
| Sn | - | - | - | - | - | - | - | - |
| Sb | - | - | - | - | - | - | - | - |
| Cs | 0.91 | 0.01 | 0.93 | 0.05 | 0.91 | 0.07 | 0.90 | 0.01 |
| Ba | 876.95 | 0.72 | 955.40 | 16.64 | 902.53 | 46.64 | 992.35 | 115.35 |
| La | 93.48 | 0.53 | 92.94 | 8.77 | 84.52 | 26.27 | 114.49 | 43.59 |
| Ce | 31.03 | 0.11 | 30.38 | 1.94 | 30.10 | 2.13 | 31.53 | 1.01 |
| Pr | 24.04 | 0.05 | 22.83 | 1.76 | 21.06 | 4.62 | 26.99 | 7.77 |
| Nd | 101.53 | 0.74 | 94.46 | 6.97 | 86.97 | 17.87 | 110.83 | 29.17 |
| Sm | 22.22 | 0.16 | 20.56 | 1.46 | 19.22 | 3.37 | 23.39 | 5.07 |
| Eu | 5.55 | 0.01 | 5.15 | 0.37 | 4.84 | 0.86 | 5.99 | 1.29 |
| Gd | 23.87 | 0.23 | 21.98 | 1.61 | 20.53 | 3.72 | 25.66 | 5.42 |
| Tb | 3.51 | 0.03 | 3.29 | 0.24 | 3.07 | 0.58 | 3.82 | 0.82 |
| Dy | 21.23 | 0.13 | 19.97 | 1.43 | 18.75 | 3.50 | 23.37 | 5.27 |
| Ho | 4.21 | 0.02 | 3.95 | 0.27 | 3.69 | 0.73 | 4.63 | 1.06 |
| Er | 11.65 | 0.13 | 10.96 | 0.77 | 10.35 | 2.06 | 12.98 | 3.12 |
| Tm | 1.60 | 0.01 | 1.53 | 0.10 | 1.43 | 0.27 | 1.78 | 0.43 |
| Yb | 9.82 | 0.04 | 9.43 | 0.60 | 8.91 | 1.76 | 10.90 | 2.56 |
| Lu | 1.38 | 0.00 | 1.33 | 0.10 | 1.25 | 0.24 | 1.53 | 0.35 |
| Hf | 0.88 | 0.01 | 0.87 | 0.07 | 0.85 | 0.07 | 0.87 | 0.02 |
| Ta | 0.11 | 0.00 | 0.11 | 0.01 | 0.11 | 0.01 | 0.11 | 0.00 |
| W | 0.00 | 0.00 | 0.00 | 0.00 | 0.00 | 0.00 | 0.00 | 0.00 |
| Tl | - | - | - | - | - | - | - | - |
| Pb | 140.39 | 0.46 | 137.75 | 6.73 | 136.71 | 10.46 | 140.02 | 2.76 |
| Bi | 1.74 | 3.02 | 1.69 | 2.92 | 0.00 | 0.00 | 0.00 | 0.00 |
| Th | 1.29 | 0.01 | 1.27 | 0.08 | 1.26 | 0.10 | 1.27 | 0.02 |
| U | 1.58 | 0.01 | 1.55 | 0.08 | 1.53 | 0.12 | 1.61 | 0.05 |
| ΣREY | 460.50 | 2.75 | 436.55 | 26.05 | 408.68 | 68.28 | 514.02 | 108.83 |

b.d.l.: below the detection limit.

## 3. Results

### 3.1. Chemical Composition of Umber Samples

Tables 1–3 show the whole rock chemical compositions of the umber samples. Although some variations were observed among the different particle size fractions, the Kunimiyama umber showed the highest ΣREY concentrations (1710–1840 ppm (mg/kg)) among the three umber samples. Fe and Mn concentrations were in the range of 25.7–28.6 wt. % (36.7–40.9 wt. % as $Fe_2O_3$*; total iron as $Fe_2O_3$) and 18.0–18.9 wt. % (23.2–24.4 wt. % as MnO), respectively. The Ca concentrations varied among the different size fractions (3.17 wt. % for 25–150 μm, 2.40 wt. % for 150–1000 μm, 1.76 wt. % for 1000–2000 μm, 1.85 wt. % for 2000–4000 μm), and it was highly likely that Ca-rich fragile minerals such as calcite were enriched in the small size fractions. The standard deviation (1δ) of REY concentrations was within 10% at the size fractions smaller than 2000 μm, but that of 2000–4000 μm reached 20%. Therefore, it should be noted that the results of leaching experiments could include this level of error.

ΣREY concentrations of the Aki umber varied widely from 450 to 771 ppm among the different size fractions. The concentrations of Fe and Mn were the lowest among the three samples and were 11.0 to 17.5 wt. % (15.7 to 25.0 wt. % as $Fe_2O_3$*) and 1.54 to 3.35 wt. % (1.99 to 4.33 wt. % as MnO), respectively. The concentrations of Fe and Mn increased with the decreasing size fraction (17.45 and 3.35 wt. % for 25–150 μm, 15.52 and 2.20 wt. % for 150–1000 μm, 10.96 and 1.54 wt. % for 1000–2000 μm, respectively) except for the 2000–4000 μm fraction in which the data uniformity was poor. A similar tendency was also observed for Na, Mg, Al, P, Ca, Ti, and REE. This was attributed to the difference in the quartz amount because quartz tends to be enriched in large size fractions given its hardness property. The standard deviation (1δ) of REY concentrations was within 7% at the size fractions smaller than 2000 μm, but that of the size fraction 2000–4000 μm reached 68.5%.

The Cyprus umber had the lowest ΣREY concentrations (408–514 ppm) and the highest Fe concentrations (36.3–38.3 wt. % as Fe and 51.9–54.7 wt. % as $Fe_2O_3$*) among the three samples. Mn showed an intermediate value (6.82–7.63 wt. %, 8.81–9.85 wt. % as MnO). The concentrations of P and Ca were 0.51–0.57 wt. % and 1.21–1.64 wt. %, respectively. This sample was compositionally homogenous, and the standard deviation (1δ) was within 7% for the average concentration at any particle size for Fe and Mn. However, the standard deviation of P reached 39% at the particle size of 2000–4000 μm. Ca showed a similar tendency. These data indicate that calcium phosphate was heterogeneously contained in this sample. The standard deviation of REY was low at the size fractions smaller than 1000 μm (0.6% for 25–150 μm, 5.97% for 150–1000 μm), and it was 16.7% for 1000–2000 μm and 21.2% for 2000–4000 μm.

### 3.2. Results of Leaching Experiments

Figures 3–8 and Supplementary Tables S1–S6 show the results of the chemical leaching experiments. In particular, Figures 3–8 show the extraction percentage of each element (major elements: Na, Mg, Al, P, K, Ca, Ti, Mn, and Fe; valuable metals: V, Co, Ni, Cu, and Mo; radioactive elements: Th and U; REE: Sc and REY) with the leaching time. REY data are subdivided into Y, Ce, light REE other than Ce (LREE: La, Pr, Nd, Pm, Sm, and Eu), and heavy REE other than Y (HREE: Gd, Tb, Dy, Ho, Er, Tm, Yb, and Lu) in these figures.

Leaching experiments for the Kuminiyama umber: The results of leaching experiments for the Kunimiyama umber are shown in Figure 3 (leachate: hydrochloric acid) and Figure 4 (sulfuric acid). In the experiments using hydrochloric acid, increasing the leaching time generally increased the extraction levels of most major elements (Na, Mg, Al, P, Ca, Ti, Mn, and Fe), valuable metals (V, Co, Ni, Cu, and Mo), radioactive elements (Th and U), and REE (Sc, Y, Ce, LREE, and HREE). Although the extraction data had a large error range reflecting the heterogeneity of samples with large particle size fractions (1000–2000 and 2000–4000 μm), the leaching behavior was similar to that of small size fractions. The extraction percentage of each element was generally higher in the smaller particle size fraction. The highest extraction percentages of REE were 40.4% for Sc, 39.5% for Y, 6.2% for Ce,

19.7% for LREE, and 34.3% for HREE under conditions of a 25–150 μm particle size with 1440 min leaching time. The extraction percentages of Fe and Mn were only 3.2% and 27.4%, respectively, at the highest (particle size: 25–150 μm, leaching time: 1440 min). On the other hand, Ca and P showed high extraction percentages up to 90% under the same condition. The results of experiments using sulfuric acid were almost the same as those for hydrochloric acid (Figure 4). However, the Ca extraction tended to decrease with leaching time at the smallest particle size fraction, which was indicative of re-adsorption and/or precipitation. In addition, the extraction percentage of Mn was relatively higher (up to 50%) than the cases using hydrochloric acid.

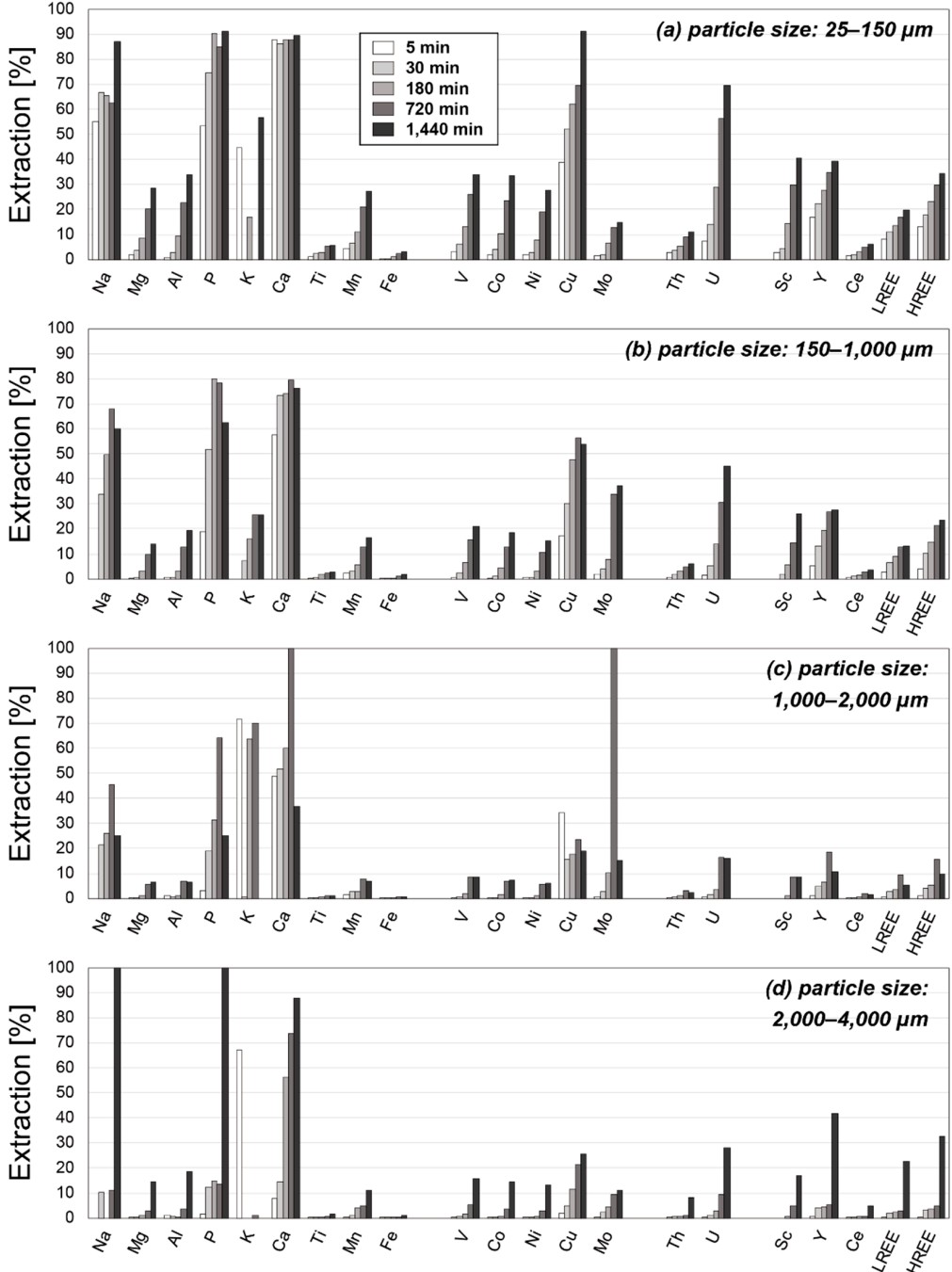

**Figure 3.** Extraction percentages of major elements, valuable metals, radioactive elements, and REE from the Kunimiyama umber as a function of time using hydrochloric acid as the leachate. Experiments with the particle size of (**a**) 25–150 μm, (**b**) 150–1000 μm, (**c**) 1000–2000 μm, and (**d**) 2000–4000 μm.

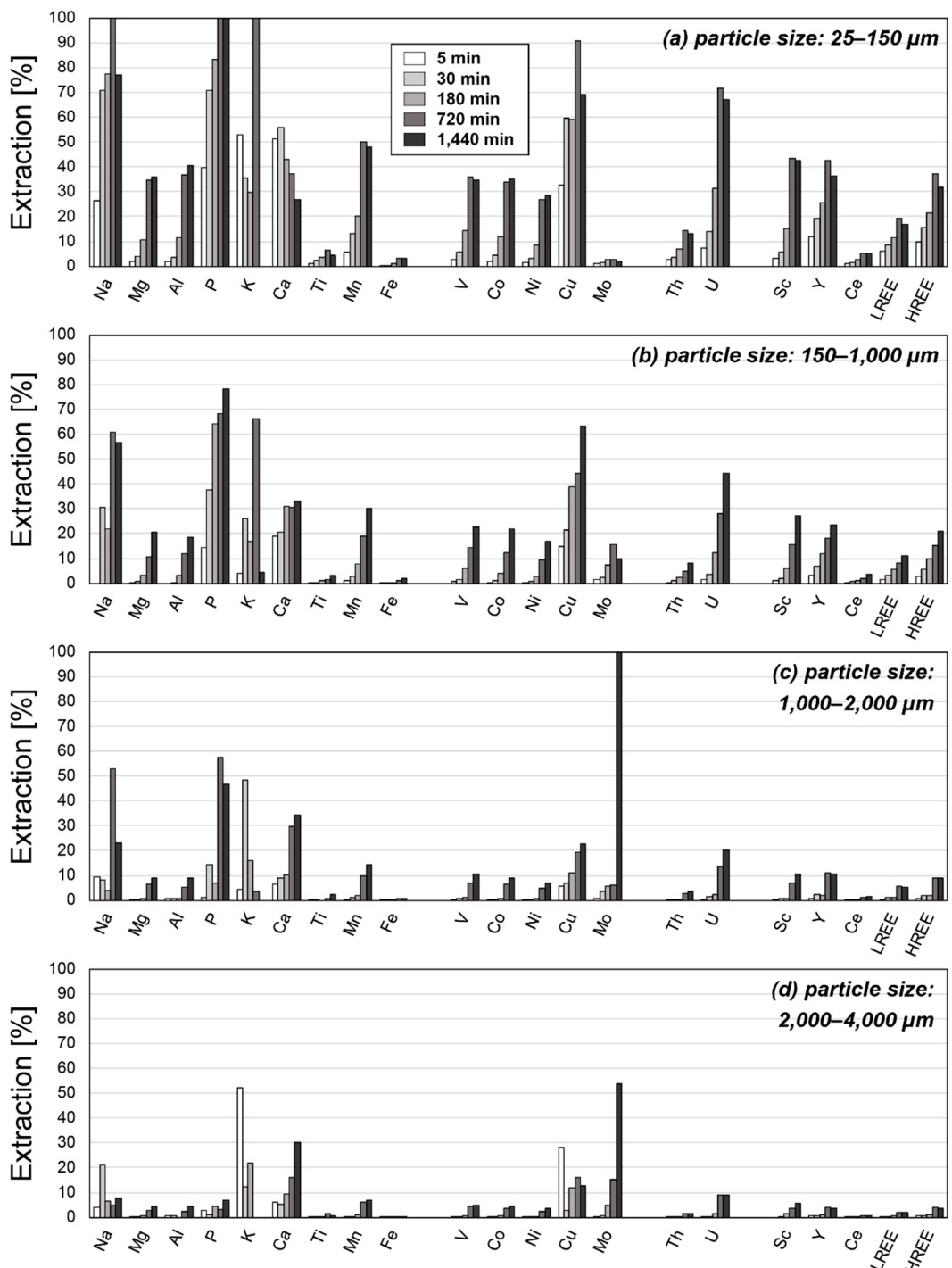

**Figure 4.** Extraction percentages of major elements, valuable metals, radioactive elements, and REE from the Kunimiyama umber as a function of time using sulfuric acid as a leachate. Experiments with the particle size of (**a**) 25–150 μm, (**b**) 150–1000 μm, (**c**) 1000–2000 μm, and (**d**) 2000–4000 μm.

Leaching experiments for the Aki umber: The results of leaching experiments for the Aki umber are shown in Figure 5 (leachate: hydrochloric acid) and Figure 6 (sulfuric acid). As with the experiments for the Kunimiyama umber, the extraction percentages of elements from the Aki umber generally increased over time with both leachates. However, Mo and Th extraction values decreased with time in some cases. Extraction percentages of REE were higher than those of the Kunimiyama umber and were 25.6% for Sc, 94.1% for Y, 42.8% for Ce, 83.8% for LREE, and 90.2% for HREE at the highest (leachate: HCl, particle size: 25–150 μm, leaching time: 720 min). The extraction percentages of Fe and Mn were

low and amounted to 2.1% and 8.6%, respectively, for hydrochloric acid (particle size: 25–150 μm, leaching time: 1440 min) and 1.9% and 8.2%, respectively, for sulfuric acid (particle size: 25–150 μm, leaching time: 1440 min) at the highest.

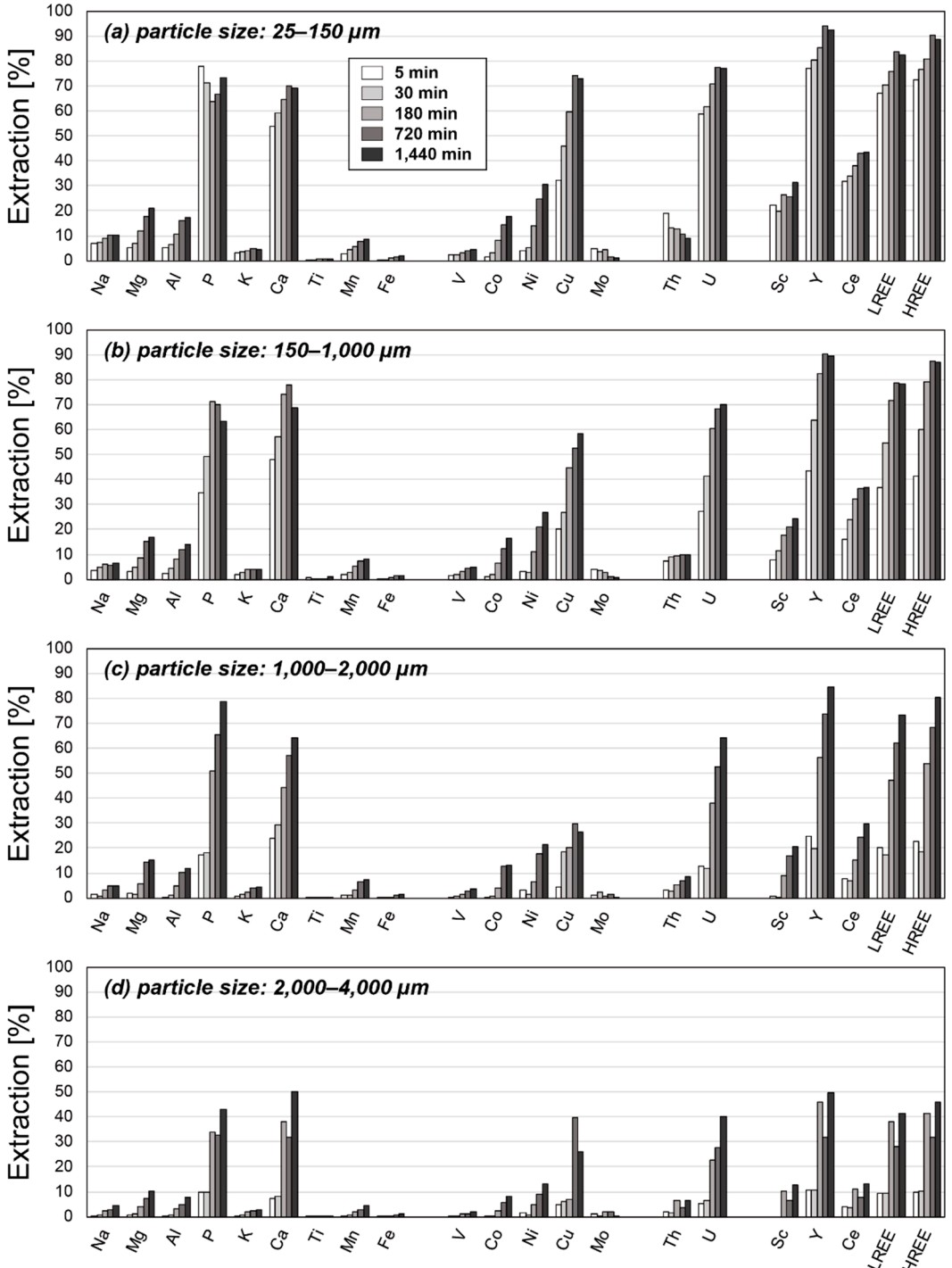

**Figure 5.** Extraction percentages of major elements, valuable metals, radioactive elements, and REE from the Aki umber as a function of time using hydrochloric acid as a leachate. Experiments with the particle size of (**a**) 25–150 μm, (**b**) 150–1000 μm, (**c**) 1000–2000 μm, and (**d**) 2000–4000 μm.

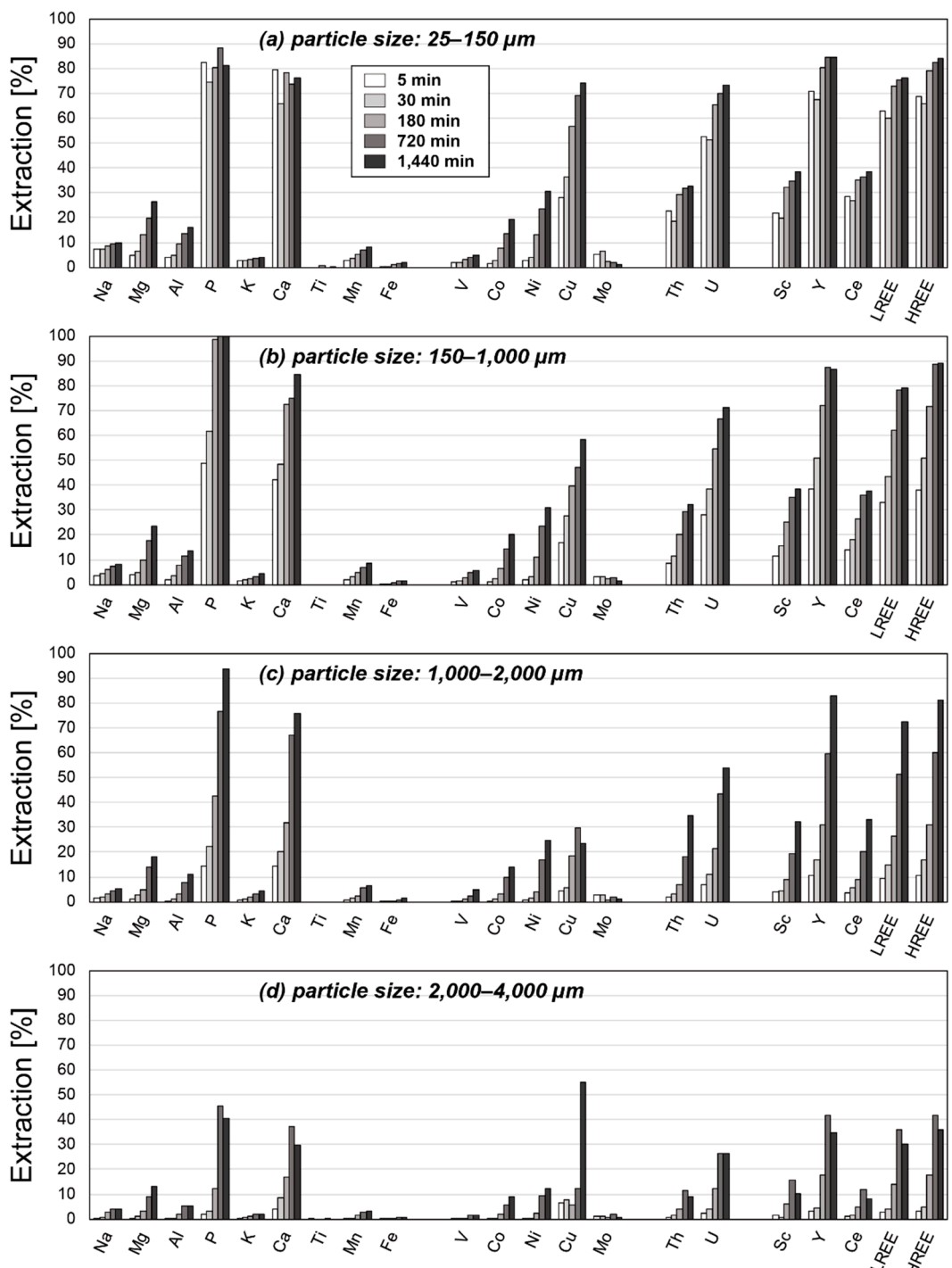

**Figure 6.** Extraction percentages of major elements, valuable metals, radioactive elements, and REE from the Aki umber as a function of time using sulfuric acid as a leachate. Experiments with the particle size of (**a**) 25–150 μm, (**b**) 150–1000 μm, (**c**) 1000–2000 μm, and (**d**) 2000–4000 μm.

Leaching experiments for the Cyprus umber: The results of leaching experiments for the Cyprus umber are shown in Figure 7 (leachate: hydrochloric acid) and Figure 8 (sulfuric acid). The extraction percentages of elements generally increased with leaching time with both leachates. However, P extraction values clearly decreased with time. REE extraction except for Sc and Ce showed high percentages up to 100% and was generally high at the large particle size fraction with both leachates. This may be because goethite at the particle surface changed to hematite during the drying and inhibited the leaching reaction by acting as a passivation layer. The extraction percentages of Fe and

Mn were low and amounted to 1.1% and 7.2%, respectively, for hydrochloric acid (particle size: 25–150 μm, leaching time: 1440 min) and 1.2% and 6.1%, respectively, for sulfuric acid (particle size: 25–150 μm, leaching time: 1440 min) at the highest. The extraction percentages of valuable metals were lower than 30% in all cases.

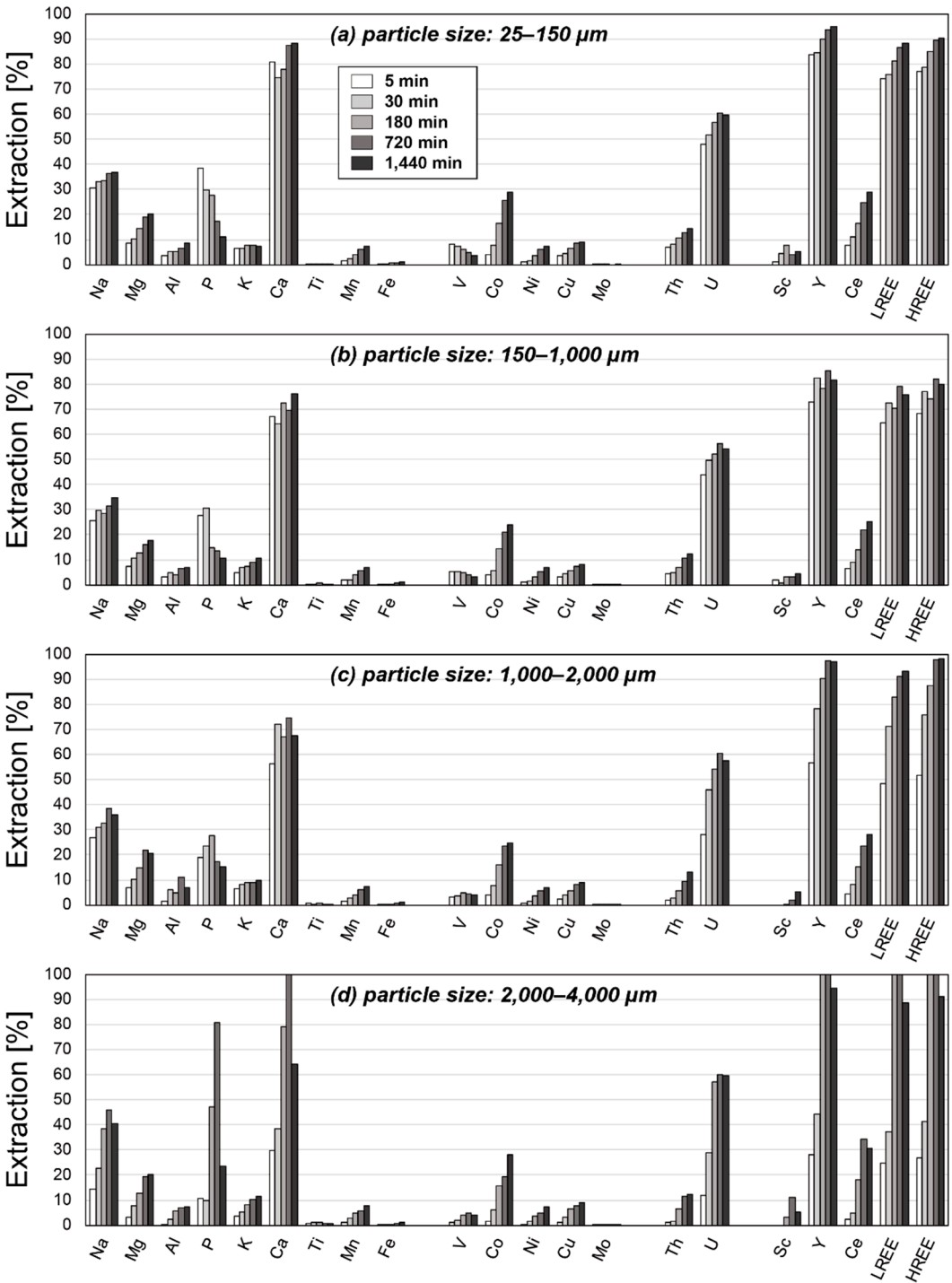

**Figure 7.** Extraction percentages of major elements, valuable metals, radioactive elements, and REE from the Cyprus umber as a function of time using hydrochloric acid as a leachate. Experiments with the particle size of (**a**) 25–150 μm, (**b**) 150–1000 μm, (**c**) 1000–2000 μm, and (**d**) 2000–4000 μm.

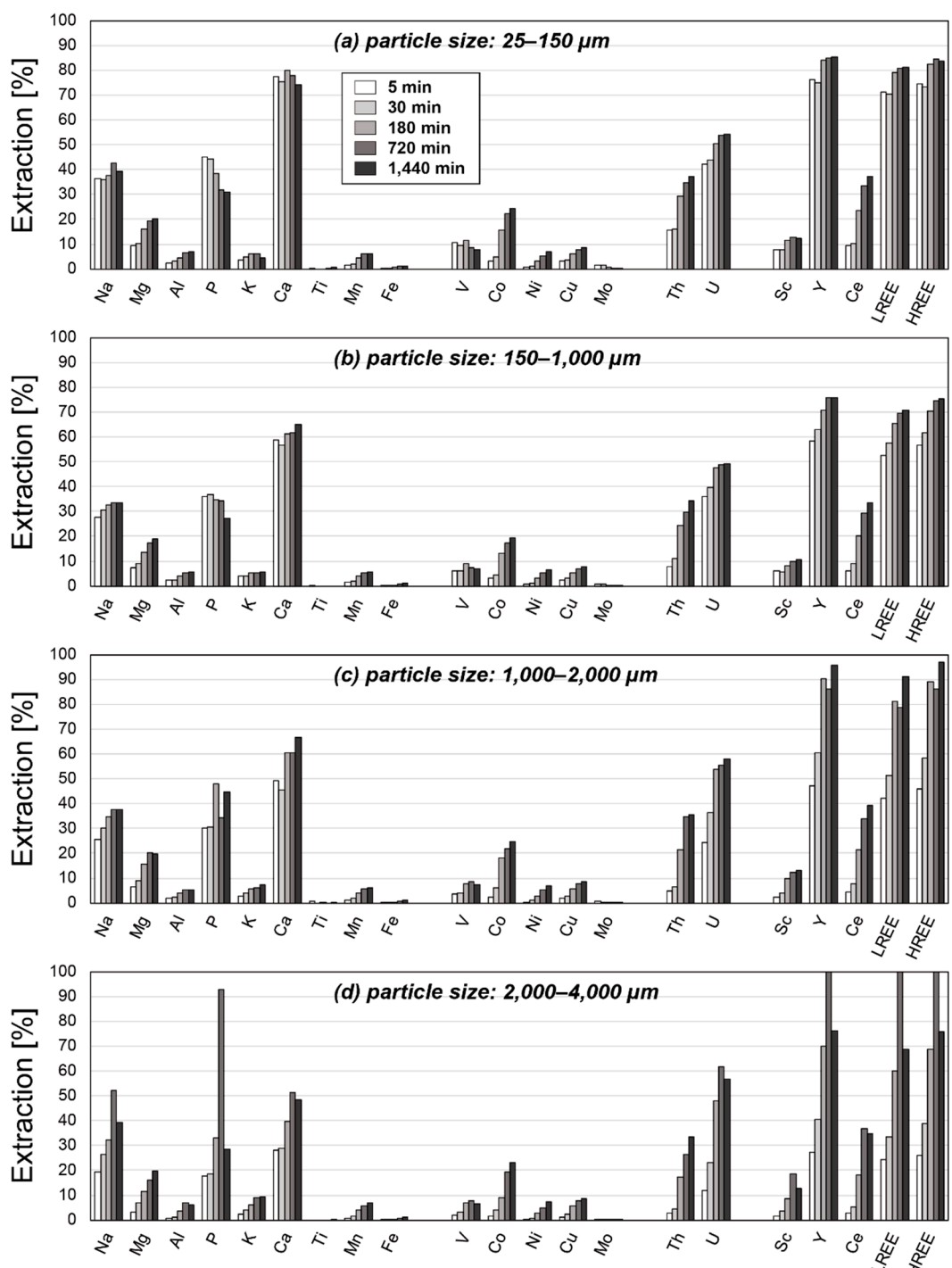

**Figure 8.** Extraction percentages of major elements, valuable metals, radioactive elements, and REE from the Cyprus umber as a function of time using sulfuric acid as a leachate. Experiments with the particle size of (**a**) 25–150 μm, (**b**) 150–1000 μm, (**c**) 1000–2000 μm, and (**d**) 2000–4000 μm.

## 4. Discussion

### 4.1. Host Minerals of REE

The experimental results for large particle size fractions were susceptible to the sample heterogeneity. In addition, it was revealed that the co-precipitation of rare-earth sulfate with calcium sulfate led to a decrease of the REY amount in the leached solution using sulfuric acid [37,38], and

thus, the dissolution behavior and the host materials of REY should be discussed based on the results of the smallest particle size fraction sample using hydrochloric acid.

In marine sediments, REY are generally hosted in calcium phosphate (apatite), Fe-Mn oxide/hydroxide, and terrigenous material [5,39–41]. We discuss the host minerals of REY in each sample on the premise that the host minerals of REY in the umber samples were the above-mentioned materials, because the umber had originated in marine sediments (hydrothermal metalliferous sediments).

In the experiments for the Kunimiyama umber, while the extraction percentage of Ca (main constituent of apatite) reached 87.8% at 5 min leaching time (Figure 3a), that of REY remained only at ~20% (Y: 16.7%, LREE: 8.3%, and HREE: 13.3%). Therefore, the amount REY in apatite was estimated to be up to 20%. On the other hand, P extraction increased with time (53.4% to 91.0% from 5 min to 1440 min) despite the observation that the Ca extraction value did not change with time (87.8% to 89.7%); therefore, the increment of P extraction from 5 to 1440 min was due to other minerals than apatite. On the other hand, the Mn extraction amount increased with time and the extraction form (changed with time) closely resembled that of REE excluding Ce (Figure 3a). These findings suggest that most of REY in the Kunimiyama umber exists with Mn. REY coexisting with the Fe and Mn oxide/hydroxide in deep-sea sediments can be extracted easily by using a long enough leaching time and by increasing the acid concentration [37,38]. This may be because REY are not incorporated into the structure of the oxide/hydroxide but are adsorbed on the mineral surface. However, the mineral phase of Mn in the Kunimiyama umber was changed to a silicate mineral (bementite) due to regional metamorphism in the process of subduction/uplift, which led to a decrease of the REY extraction. Ce extraction showed lower values than other REE. Ce is oxidized easily to the tetravalent form in the presence of oxidative seawater, and it precipitates as oxide and is removed from seawater [42,43]. Therefore, Ce is depleted compared with other REE in seawater, and it is strongly influenced by substances other than those of seawater origin (mainly terrigenous material). Terrigenous material is mainly composed of silicate minerals, which generally have a high acid tolerance, and therefore, Ce extraction from umber samples showed low values. As with Ce, Ti also showed low extraction percentages.

In the experiments for the Aki umber, the extraction percentages of REY were significantly higher than those of the Kunimiyama umber (Figure 5a). Y, LREE, and HREE were extracted to the extent of 84–94% at 720 min of leaching time. Ca and P extraction also reached 70% and 67%, respectively, at 720 min, and their extraction form resembled that of REE excluding Sc and Ce. On the other hand, the extraction of Fe and Mn remained low, at 1.7% and 7.9% at 720 min, respectively. These dissolution behaviors indicate that the main host mineral of REY in the Aki umber is apatite. Since apatite easily dissolves in acid solution rather than other silicate minerals, the dissolution of apatite, and the extraction of REY, preferentially proceeded in our experiments. Because the extraction percentages of Ca and P reached up to about 70%, it can be concluded that Ca and P are also contained in minerals other than apatite, such as silicate and Fe-Mn oxide/hydroxide.

The Cyprus umber showed the highest REY extraction among the three samples (Figure 7a), and the extraction percentages of Y and HREE exceeded 90%. Ca extraction also reached 80% at 5 min. In addition, the extraction form of Ca in the experiments with the large particle size fraction (2000–4000 μm) resembled that of REE excluding Sc and Ce (Figure 7d). This result strongly suggests that the main host mineral of REY in the Cyprus umber is apatite. While Ca showed high extraction levels, the extraction percentages of P, the main constituent element of apatite, were low compared to those of Ca. Thus, in this umber sample, some amount of P is also contained in Fe hydroxide (goethite) as well as apatite. However, goethite did not dissolve in our experiments and the extraction of Fe remained only at 1.1% even at 1440 min. Since it is well-known that Fe-Mn oxide/hydroxide easily adsorbs many elements [10–12], the decrease of P extraction (from 38.7% at 5 min of leaching time to 10.9% at 1440 min) was possibly due to the re-absorption of extracted P onto goethite.

### 4.2. Factors Affecting the Extraction Behaviors of REE

In our experiments, the three umber samples showed different leaching behaviors and different extraction percentages were obtained for the target elements. These differences are attributable to the different chemical, mineral, and physical properties of the samples.

Our samples had a wide range of chemical compositions and consisted of different mineral phases. As described above, the host minerals of REY were different among the three samples and this affected the extraction percentages. However, it was revealed that the physical properties of umber samples essentially controlled REY extraction. We measured the bulk density and P-wave propagation velocity of the samples to examine the effects of physical properties on the leaching behaviors. Bulk density was calculated by measuring the weight of 1–1.5 cm cube samples. P-wave propagation velocity was measured by using Pundit (C.N.S. Electronics Ltd., London, UK), a portable test device. Generally, the bulk density and P-wave propagation velocity of the umber samples showed a good correlation with the porosity and/or permeability, which strongly affected the leaching behaviors of the umber samples. Figure 9a demonstrates how the extraction percentages of REY decreased with the increases in the density and P-wave propagation velocity of umber samples. This figure also shows that the extraction of REY from different umber samples can be predicted based on the density and P-wave propagation velocity data regardless of the differences in the chemical and/or mineral composition.

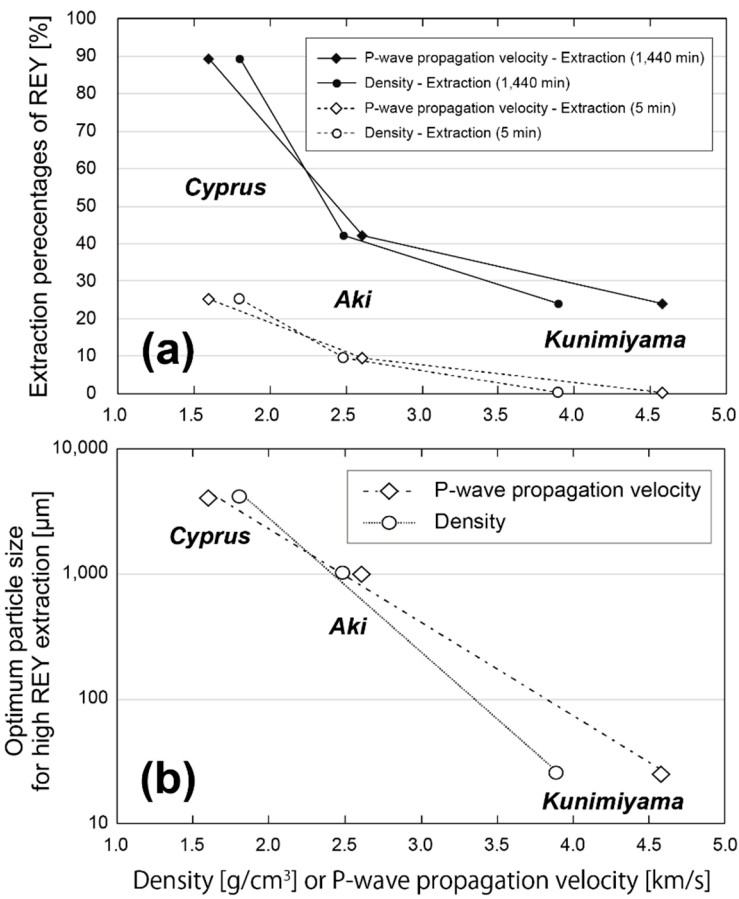

**Figure 9.** Effect of physical properties (density and P-wave propagation velocity) of the umber samples on the (**a**) extraction percentages of REY and (**b**) optimum particle size for high REY extraction.

The differences in the physical properties of the umber samples can be attributed to the pressure and thermal history of each sample. The umber originated from marine sediments (hydrothermal metalliferous sediments) and, therefore, accreted on land through the geological processes of subduction/obduction. Under the microscope, the constituent minerals of the Kunimiyama greenstone,

which is a wallrock of the Kunimiyama umber, were determined to be albite, chlorite, calcite, epidote, pumpellyite, prehnite, quartz, celadonite, and sericite with opaque minerals of Fe-oxyhydroxide and hematite [17]. Plagioclase was completely replaced by albite and only a relict of clinopyroxene was observed as the primary igneous mineral. The metamorphic mineral assemblages of epidote + pumpellyite + chlorite and chlorite + pumpellyite + prehnite are indicative of the metamorphic condition of typical prehnite-pumpellyite facies [17]. In the Aki greenstone, primary plagioclase often remained through the metamorphism with the primary clinopyroxene phenocryst. The Aki greenstone has a mineral assemblage of albite (plagioclase), clinopyroxene, chlorite, calcite, and quartz with an opaque mineral. Because no epidote was observed and primary plagioclase often remained, it can be concluded that the Aki umber underwent weaker metamorphism than the Kunimiyama umber. On the other hand, the Cyprus umber abducted on land (the Anatolian Plate) as a part of the Troodos ophiolite and did not undergo regional metamorphism at the subduction zone except for seafloor weathering. Thus, the differences in the degree of metamorphism controlled the physical properties of the three umbers and their REY extraction behaviors.

### 4.3. Optimum Conditions for REY Extraction from Umber Samples

Optimum acid leachate: In our experiments, REY extraction with sulfuric acid was slightly lower than that with hydrochloric acid, possibly because of the co-precipitation of REE-sulfate with Ca-sulfate and the precipitation of REE-Na double sulfate [38]. However, the differences were slight for all samples. REY extraction from the smallest particle size fraction at 1440 min leaching time was 21.9% with hydrochloric acid and 19.0% with sulfuric acid for the Kunimiyama umber, 83.6% and 77.5%, respectively, for the Aki umber, and 88.5% and 81.5%, respectively, for the Cyprus umber. Importantly, industrial sulfuric acid is less expensive to use than hydrochloric acid [44]. Specifically, the costs of using sulfuric acid are about one fifth of those of using hydrochloric acid to make same the normality acid leachate. Therefore, sulfuric acid is economically preferable for the leachate of umber.

Optimum grinding level: The extraction of REY from the Kunimiyama umber reached only 21.9% even with the smallest particle fraction. On the other hand, REY extraction exceeded 80% for the Aki umber with the 150–1000 μm fractions (80.2% at 720 min) and for the Cyprus umber with the 2000–4000 μm (100% at 180 min) fraction. Figure 9b compares the physical properties (density and P-wave propagation velocity) to the particle size, which helped to ensure high REY extraction (>80%). The optimum particle size of each sample for high REY extraction was assumed to be 25 μm (Kunimiyama), 1000 μm (Aki), and 4000 μm (Cyprus) based on the experimental results. In Figure 9b, the optimum particle size for high REY extraction increased exponentially with the increase of rock density and P-wave propagation velocity, and therefore, samples showing low bulk density and P-wave propagation velocity will have lower grinding costs and are economically advantageous. As described above, the Cyprus umber (Troodos ophiolite) was exposed on land by the obduction process and did not undergo regional metamorphism. On the other hand, umber samples from Japanese accretionary complexes, especially the Kunimiyama umber, have underwent regional metamorphism up to prehnite-pumpellyite facies during the subduction process, thus resulting in dense textures and low REY extraction potential. Our results (Figure 9b) show that the optimum (requisite) particle size for high REY extraction is easily predictable based on the physical properties of rock samples.

## 5. Conclusions

In this study, a series of chemical leaching experiments with three different umber samples (Kunimiyama, Aki, and Cyprus) were conducted to elucidate the controlling factors of REY extraction and the optimum leaching conditions. We prepared four size fractions (25–150, 150–1000, 1000–2000, and 2000–4000 μm) for each umber sample, which were used in the experiments to examine the optimum grinding level. The umber samples showed a wide range of chemical compositions and physical properties. Our results clearly demonstrated that REY extraction (extraction percentages and/or optimum particle size for high REY extraction) from the umber samples is predictable simply

based on the samples' physical properties (density and P-wave propagation velocity). In our samples, the Kunimiyama umber, which underwent regional metamorphism up to prehnite-pumpellyite facies during the subduction process and had a high density and P-wave propagation velocity, showed the lowest REY extraction (~21.9%). On the other hand, the Cyprus umber, which did not undergo metamorphism except for seafloor weathering and had a low density and P-wave propagation velocity, showed the highest REY extraction (~100%). It is necessary to consider the economic value (concentrations of REY) for feasibility analyses, however our results strongly indicate that umber samples that underwent strong metamorphism are not suitable for actual development because of the dense texture. In addition, our experiments showed that sulfuric acid is a more economical leachate for chemical leaching of umber samples.

**Supplementary Materials:** The following are available online at http://www.mdpi.com/2075-163X/9/4/239/s1, Tables S1–S6. Table S1: Extraction amounts of major elements, trace elements, and rare earth elements from the Kunimiyama umber using hydrochloric acid as a leachate. Table S2: Extraction amounts of major elements, trace elements, and rare earth elements from the Kunimiyama umber using sulfuric acid as a leachate. Table S3: Extraction amounts of major elements, trace elements, and rare earth elements from the Aki umber using hydrochloric acid as a leachate. Table S4: Extraction amounts of major elements, trace elements, and rare earth elements from the Aki umber using sulfuric acid as a leachate. Table S5: Extraction amounts of major elements, trace elements, and rare earth elements from the Cyprus umber using hydrochloric acid as a leachate. Table S6: Extraction amounts of major elements, trace elements, and rare earth elements from the Cyprus umber using sulfuric acid as a leachate.

**Author Contributions:** Y.T., K.F., E.U., and Y.K. designed the study. N.T. and K.F. conducted the field investigations and sampling. Y.T. and M.W. conducted the leaching experiments and chemical analyses. Y.T., M.W., and E.U. conducted the physical property measurements. All authors participated in the discussion to interpret the results, and Y.T. wrote the paper.

**Funding:** This work was financially supported by the Japan Society for the Promotion of Science (JSPS) KAKENHI Grant No. 17K06985 to K.F. and Y.T.

**Acknowledgments:** We thank the editor and two anonymous reviewers for their constructive comments.

**Conflicts of Interest:** The authors declare no conflict of interest.

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
