# Peer review of "Experiments on Rare-Earth Element Extractions from Umber Ores for Optimizing the Grinding Process"

_minerals, doi:10.3390/min9040239_

Round 1

Reviewer 1 Report

Do not recall words from the title in the keywords

In the introduction first paragraph, also consider the work from French and British researchers, check the papers from Josso et al. 2017 and Menendez et al. 2017 in a special issue of ore geology review edited by James Hein. Also check the Elements issue from last November.

On Fig 1 write correctly Goethite.

L127 do not use the obsolete unit N, check NIST recommendation 

Replace ppm by SI unit mg/kg.

Authors need to further discussed the difference with Josso et al 2018 paper [9].

Author Response

Response to the comments from Reviewer 1

Dear Sir/Madam:

We wish to express our appreciation for your helpful comments on our paper. We have read your comments/suggestions carefully and revised our manuscript accordingly. The following material presents our replies (in blue) to your specific comments (in black):

1) Do not recall words from the title in the keywords

In accordance with your comment, we have modified the keywords as follows;

ferromanganese deposits; metalliferous sediment; chemical leaching; REY; accretionary complex

2) In the introduction first paragraph, also consider the work from French and British researchers, check the papers from Josso et al. 2017 and Menendez et al. 2017 in a special issue of ore geology review edited by James Hein. Also check the Elements issue from last November.

6) Authors need to further discussed the difference with Josso et al 2018 paper [9].

In accordance with your comment, we have modified the last paragraph of the Introduction to explain the difference with the previous studies (especially Josso et al. [9]) as follows:

There are many umber deposits in Japanese accretionary complexes, and resource amount estimations as well as geochemical studies already have been conducted [7,8,18–21]. Umber deposits are also extensively exposed on land as part of ophiolites such as in Cyprus (Troodos ophiolite) and Oman (Semail ophiolite) [22–24]. Josso et al. [9] conducted REY extraction experiments on the Cyprus umber and discussed the optimum procedural conditions (acid concentration, temperature, solid-liquid ratio, and leaching time). However, the factors that control the leaching behavior of umber were not clarified in the previous studies. In this study, therefore, we conducted a series of chemical leaching experiments using three different umber samples collected from Japanese accretionary complexes and Troodos ophiolite to elucidate the factors affecting the extraction behaviors of REY. In addition, we have established a simple prediction method for REY extraction efficiency from umber ores targeted for future development.

3) On Fig 1 write correctly Goethite.

We have corrected Figure 1.

4) L127 do not use the obsolete unit N, check NIST recommendation

We have modified the text as follows:

The acid concentration was set to 0.5 mol/L for hydrochloric acid and 0.25 mol/L for sulfuric acid, and the leaching temperature was set to 25°C.

5) Replace ppm by SI unit mg/kg.

The [ppm] unit is commonly used in the papers in the field of earth sciences and engineering; therefore, we did not change this notation. However, in accordance with your comment, we also used the SI unit [mg/kg] along with [ppm] in the first appearance of [ppm] in the manuscript.

Again, we deeply thank you for your helpful comments.

Sincerely yours,

Yutaro Takaya

Reviewer 2 Report

There is a reasonable explanation in the introduction, but it could be improved by noting past work in this space to equal that of a existing basic geological description.  

Please check grammar -comprises, rather than composes... (Line 67 and one other)

It would be prudent to disclose the equipment set up - photo/schematic/ whatever...

Name the brand of the XRD and software used in this study.

Please check through how you describe the material fractions, particularly the tables.  Particles are made from broken rock.  Particles comprise grains of individual minerals.  "Fines" are removed from the particles using ultra sonic cleaning.  Make sure that the tables are talking about particle sizes and not grains - where applicable.

Line 132 - What is a PFA vessel?  Show/explain.

Line 202 Precipitation/passivation - Do you categorically know that this is the case?  If the rate of reactions slows, is it passivation or not?  Did you look at surface chemistry?  XPF technique may shed light on this.   As with most ores and grinding/leaching, the smaller the particles, the greater the surface to volume ratio and the "easier" it is to leach.  However if passivation does occur is this still the case?  Comments related to this would be interesting.

Grade recovery curves would be interesting in this space.  Recovery/time.

A brief mention of costs/economics is not really the focus of this paper, may rethink whether to include that or not.  

There is room to put more of your text into relevant graphs which would in my opinion add value to the paper.

It is interesting and worth spending a little more time considering illustrating the results and adding images if this is possible.

Author Response

Response to the comments from Reviewer 2

Dear Sir/Madam:

We greatly appreciate your insightful comments on our paper. The comments have helped us significantly improve the paper. We have incorporated all your comments/suggestions in our revised manuscript.

The following material contains our reply (in blue) to your specific comments (in black):

1) There is a reasonable explanation in the introduction, but it could be improved by noting past work in this space to equal that of a existing basic geological description.

In accordance with your comment, we have modified the last paragraph of the Introduction to explain the difference with the previous studies (especially Josso et al. [9]) as follows:

There are many umber deposits in Japanese accretionary complexes, and resource amount estimations as well as geochemical studies already have been conducted [7,8,18–21]. Umber deposits are also extensively exposed on land as part of ophiolites such as in Cyprus (Troodos ophiolite) and Oman (Semail ophiolite) [22–24]. Josso et al. [9] conducted REY extraction experiments on the Cyprus umber and discussed the optimum procedural conditions (acid concentration, temperature, solid-liquid ratio, and leaching time). However, the factors that control the leaching behavior of umber were not clarified in the previous studies. In this study, therefore, we conducted a series of chemical leaching experiments using three different umber samples collected from Japanese accretionary complexes and Troodos ophiolite to elucidate the factors affecting the extraction behaviors of REY. In addition, we have established a simple prediction method for REY extraction efficiency from umber ores targeted for future development.

2) Please check grammar -comprises, rather than composes... (Line 67 and one other)

The revised manuscript has undergone grammar checking by a professional English editing service and slight modifications were made including to this part.

3) It would be prudent to disclose the equipment set up - photo/schematic/ whatever...

In accordance with your comment, we have added a schematic figure of the experimental procedure (Figure 2).

4) Name the brand of the XRD and software used in this study.

We have added the name of the XRD and software in Section 2.1.

5) Please check through how you describe the material fractions, particularly the tables. Particles are made from broken rock. Particles comprise grains of individual minerals. "Fines" are removed from the particles using ultra sonic cleaning. Make sure that the tables are talking about particle sizes and not grains - where applicable.

In the original manuscript, we confused “grain” with “particle.” Therefore, in accordance with your comment, we have modified the term “grain” to “particle” in the text, tables, and figures.

6) Line 132 - What is a PFA vessel? Show/explain.

We have modified the description of the vessel as follows:

A sample weighing 0.2 ± 0.002 g was combined with 3 mL of the leachate in a PTFE (polytetrafluoroethylene) vessel.

7) Line 202 Precipitation/passivation - Do you categorically know that this is the case? If the rate of reactions slows, is it passivation or not? Did you look at surface chemistry? XPF technique may shed light on this. As with most ores and grinding/leaching, the smaller the particles, the greater the surface to volume ratio and the "easier" it is to leach. However if passivation does occur is this still the case? Comments related to this would be interesting.

In accordance with your comment, we re-examined the experimental results, especially in regard to the Cyprus umber. Cyprus umber showed strange leaching behavior in that the REY recovery increased with the increase of particle size. This strange phenomenon could be attributable to the surface chemistry of the umber sample. The main constituent mineral of the Cyprus umber is goethite. Goethite easily changes to hematite under dry conditions and even under low temperatures (<100°C) (Guo and Barnard, 2011). In our experiment, the sieved samples were dried at 60°C before the leaching experiments. In this procedure, some amount of goethite near the particle surface likely changed to hematite and inhibited the REY leaching (possibly acting as a passivation layer). This effect may have been larger in the smaller particle size fraction, and therefore, the REY extraction level of the smaller particle size fraction was lower than that of the larger fraction.

In the revised manuscript, we have added the following text to mention the above phenomena:

REE extraction except for Sc and Ce showed high percentages up to 100% and was generally high at the large particle size fraction with both leachates. This may be because goethite at the particle surface changed to hematite during the drying and inhibited the leaching reaction by acting as a passivation layer.

8) Grade recovery curves would be interesting in this space. Recovery/time.

We already discussed the recovery change with time by using Figures 2–7 (Figures 3–8 in the revised manuscript), which show the recovery/time change of each element (including rare earth elements).

9) A brief mention of costs/economics is not really the focus of this paper, may rethink whether to include that or not.

We removed the following text in the revised manuscript:

For example, sulfuric acid costs 1.5 USD (160 JPY), whereas hydrochloric costs 6.8 USD (750 JPY) to make 1 ton and 0.5 N acid leachate.

10) There is room to put more of your text into relevant graphs which would in my opinion add value to the paper. It is interesting and worth spending a little more time considering illustrating the results and adding images if this is possible.

In accordance with your comment, we have added a figure (Figure 2) and explanation of the experimental results as mentioned in our response to comment number 7.

Again, we deeply thank you for your helpful comments.

Sincerely yours,

Yutaro Takaya
